# Not All Asians are the Same: A Disaggregated Approach to Identifying Anti-Asian Racism in Social Media

## ABSTRACT

Recent policy initiatives have acknowledged the importance of disaggregating data pertaining to diverse Asian ethnic communities to gain a more comprehensive understanding of their current status and to improve their overall well-being. However, research on anti-Asian racism has thus far fallen short of properly incorporating data disaggregation practices. Our study addresses this gap by collecting 12-month-long data from X (formerly known as Twitter) that contain diverse sub-ethnic group representations within Asian communities. In this dataset, we break down anti-Asian toxic messages based on both temporal and ethnic factors and conduct a series of comparative analyses of toxic messages, targeting different ethnic groups. Using temporal persistence analysis, *n*-gram-based correspondence analysis, and topic modeling, this study provides compelling evidence that anti-Asian messages comprise various distinctive narratives. Certain messages targeting sub-ethnic Asian groups entail different topics that distinguish them from those targeting Asians in a generic manner or those aimed at major ethnic groups, such as Chinese and Indian. By introducing several techniques that facilitate comparisons of online anti-Asian hate towards diverse ethnic communities, this study highlights the importance of taking a nuanced and disaggregated approach for understanding racial hatred to formulate effective mitigation strategies.

## CCS CONCEPTS

• **General and reference** → **General conference proceedings**; • **Social and professional topics** → **Race and ethnicity**; • **Networks** → *Social media networks*.

## KEYWORDS

Anti-Asian sentiment, Racism against Asian, Panethnicity, Disaggregated Asian American data, Topic modeling, Social media mining

**ACM Reference Format:**
Anonymous Author(s). 2018. Not All Asians are the Same: A Disaggregated Approach to Identifying Anti-Asian Racism in Social Media. In *Proceedings of Make sure to enter the correct conference title from your rights confirmation emai (Conference acronym 'XX).* ACM, New York, NY, USA, 17 pages. https://doi.org/XXXXXXX.XXXXXXX

## 1 INTRODUCTION

In 2023, the U.S. government released its inaugural report of the White House Initiative on Asian Americans, Native Hawaiians, and Pacific Islanders (WHIAANHPI) [40], which aims to develop strategies to enhance justice, equity, and the overall well-being of this population (collectively referred to as Asians hereafter). One of the key priorities of this initiative is to "make disaggregated data collection and reporting the norm" across the federal agencies (WHIAANHI, [40, p.22]). Given the diverse range of ethnic groups within the Asian American population, the use of disaggregated data practices is imperative for attaining a thorough understanding of these distinct Asian communities and relevant policy-making [44]. For example, when information is reported in an aggregated manner, the average cancer rate for Asian women is lower than that for white women. However, when examining segmented records, it becomes evident that Laotian women have cancer rates more than nine times higher than those for white women (WHIAANHI, [40, p.22]). This difference highlights the critical need for disaggregated data, as it reveals the significant disparities within the Asian American population, enabling policymakers to develop targeted and effective interventions for specific communities like Laotian women. Indeed, the importance of collecting and reporting disaggregated data extends beyond Asian Americans and should be applied to all "panethnic" communities worldwide [29].

Addressing anti-Asian hate can also benefit from disaggregated data practices. Research on anti-Asian hate has attracted significant attention, especially in response to the surge in Sinophobia, a fear or dislike of China or its people, and hate crimes targeting Asians in the midst of the COVID-19 pandemic. Negative sentiments towards China and Chinese, as evidenced by derogatory labels such as "Chinese virus," along with implicit biases against Asians, have increased during the pandemic [6, 39, 46]. Federal law enforcement agencies in the U.S. have alerted the surge in anti-Asian hate crimes during this period [23]. Various advocacy efforts, including hashtag campaigns such as "#racismisvirus" and "#stopAsianhate" have also emerged to counter such anti-Asian sentiments and hate crimes.

As a result, the majority of recent studies on anti-Asian hate have utilized datasets pertaining to the influence of the COVID-19 pandemic, focusing on the evidence and consequences of Sinophobia [35, 36, 38]. While the pandemic has undoubtedly served as an important backdrop for recent Asian hate research, existing literature has failed to fully acknowledge the problem of anti-Asian sentiments as an enduring social issue that transcends being merely a byproduct of the pandemic. Furthermore, it does not adequately acknowledge that the problem of anti-Asian hate affects a wide range of ethnic groups within Asian populations, extending beyond the Chinese community.

The purpose of this study is to fill this void by examining online anti-Asian hate using a disaggregated-data approach. In particular, this study broadens the observation period to cover an extended

time frame that encompasses the pre-pandemic, peak pandemic, and post-pandemic phases, and conducts comparative analyses using disaggregated data based on both temporal and sub-ethnic breakdowns. This disaggregated approach enables the identification of nuanced distinctions in the animosity directed toward different ethnic groups within Asian populations. Moreover, it facilitates a deeper understanding of the intricate inter-ethnic dynamics within pan-Asian communities.[1]

The study aims to contribute to the literature by (1) creating a longitudinal multi-ethnic Asian hate dataset, (2) investigating temporal trends of anti-Asian messages on X (formerly known as Twitter), and (3) introducing techniques that enable comparisons of anti-Asian topics across multiple ethnic communities within pan-Asian populations. The empirical results presented in this paper address the following research questions.

(1) **RQ1:** (a) Are there changes in the magnitude of anti-Asian messages over time? (b) How do the trends over time vary across different ethnic groups?

(2) **RQ2:** (a) How semantically distant are anti-Asian messages when comparing those aimed at Asians in a general sense to those directed at specific sub-ethnic groups? (b) How do the semantic distances change over time?

(3) **RQ3:** (a) How are these topics distributed among messages targeting Asians in a general sense, those targeting major ethnic groups like Chinese and Indian, and those directed at smaller ethnic groups? (b) What are the prevalent topics of anti-Asian messages?

We collect a 12-month-long social conversations on X (formerly known as Twitter) that contain diverse sub-ethnic group representations within Asian communities. Using this dataset, we disaggregate anti-Asian toxic messages based on temporal and ethnic breakdowns and conduct a series of comparative analyses of toxic messages targeting various ethnic groups.

Findings from temporal persistence analysis, *n*-gram-based correspondence analysis, and topic modeling reveal several key insights. First, there is a substantial increase in the number of anti-Asian messages (especially anti-Chinese) in response to the declaration of the pandemic, but the average toxicity score has not much affected by the pandemic. Second, results align with previous research focused on online hatred towards the Chinese ethnicity, highlighting that toxic messages, broadly referring to 'Asians', had more semantic similarities with those targeting the Chinese ethnicity than messages aimed at other specific groups within the Asian community and that the volume of messages targeting other sub-Asian ethnic groups was relatively low. Third, *n*-gram-based analysis shows that toxic messages that attack minority ethnic groups display orthogonal semantic features compared to majority-ethnicity-attacking (e.g., Chinese, Indian) or generic-Asian-attacking messages. In contrast, when analyzing minority ethnic groups collectively using topic modeling, generic-Asian-attacking messages demonstrate more similar narrative patterns to the collective set of minority Asian ethnic groups than to a single large group such as Chinese or Indian.

In essence, this study underscores the importance of recognizing and addressing the diversity of anti-Asian hate speech. Online anti-Asian hate speech is complex and nuanced, encompassing various ethnic backgrounds and the intricate web of biases that exist both within and beyond the Asian community. In this sense, a multifaceted and disaggregated data approach is necessary to understand and combat the hateful discourse. The methodological approaches we develop in this paper may be useful to researchers and policymakers striving to better comprehend and confront these pressing challenges, fostering a more inclusive and equitable digital landscape for all. Importantly, while the primary focus of this study is on Asians, "panethnicity" is a form of identification observed globally, encompassing communities like Latino, Yoruba, or Roma [29]. Therefore, disaggregated data practices have universal applicability in addressing social issues relevant to panethnic communities.

## 2 RELATED WORK AND PROBLEM STATEMENT

### 2.1 Online Hate/toxic Speech Research

Hate and toxic speech involves abusive and aggressive language that attacks a person or group based on attributes such as race, religion, ethnic origin, national origin, sex, disability, sexual orientation, or gender identity [4, 11, 20, 34]. Much effort in this research domain has been put on message discovery solutions based on natural language techniques and models to detect and classify hate speeches more efficiently [30, 31, 37, 45]. Especially, deep learning has emerged as a powerful technique that learns hidden data representations and achieves better performance in detecting online hate speech [20, 33]. As a computational aide, state-of-the-art deep learning models such as BERT[2], a BERT fine-tuning model, RoBERTa [22] have been extensively employed [10, 31].

### 2.2 Online Anti-Asian Hate Speech Research

Anti-Asian hate speech has recently received attention in response to the outbreak of COVID-19, during which racism and hateful messages against Asians have become rampant [12, 17, 20, 47]. Online anti-Asian hate speech research has evolved into four types—COVID-specific hate speech, general anti-Asian sentiments, anti-Chinese political sentiments, and counter-hate movements such as "#racismisvirus" and "#stopAsianhate" [21]. Like previous studies on racist hate speech, anti-Asian speech research has focused on detecting and classifying anti-Asian toxic contents [20, 21, 43]. Most of these studies have centered specifically on the COVID-19 pandemic. For example, a study introduced a new classifier that identifies and categorizes online anti-Asian tweets during COVID-19 into four classes: hostility against East Asia, criticism of East Asia, meta-discussions of East Asian prejudice, and a neutral class [43]. Several studies have focused on the trends and features of anti-Asian sentiment during COVID-19 [12, 19, 27] and found that antipathy against Chinese had spillover effects on Asians in general [28]. One study uses a large-scale web-based media database to compare global sentiments toward Asians across 20 countries before and after the pandemic, finding that even though anti-Asian

---

[1]https://www.pewresearch.org/race-ethnicity/2022/08/02/what-it-means-to-be-asian-in-america/

[2]Bidirectional Encoder Representations from Transformers [7]

sentiments are deep-seated and predicated on structural undercurrents of culture, the pandemic has indirectly and inadvertently exacerbated those anti-Asian sentiments [27].

## 2.3 Filling the Void: Considering Temporal and Ethnic Heterogeneity in Asian Hate Speech

While existing research has developed various statistical/machine learning (ML) techniques (e.g., hate speech detection) to identify patterns in anti-Asian sentiments of online speech, the vast amount of research has been situated in a specific empirical context, that is, the COVID-19 pandemic, resulting in a rather skewed research trend. Although COVID-19 has resurfaced the concerns about anti-Asian hate, anti-Asian racism has been an enduring problem of inter-ethnic relations. Furthermore, empirical datasets related to COVID-19 often feature a disproportionately large number of messages concerning China and Chinese, leading to an assessment of anti-Asian sentiments that is centered around Chinese-related contents [35, 36, 38]. Even many studies, which examine a generically-defined 'Asians', have (misleadingly) alluded to Asians as being a homogeneous unity, dismissing the essence of "panethnicity" [29] that Asian is a concept that bridges very diverse sub-ethnic groups.

While those statistical/ML methods have gained traction as a pragmatic solution to mitigate the discursive "pollution" in digital information commons [26], critics point out that such models often miss contextual nuances, such as bias in different demographic and psycho-graphic subgroups [13]. Some researchers have call for a more proactive mitigation strategy beyond automated detection. For example, one study suggested that the polarized opinions sentiment analyzer system can be used as a plug-in by Twitter to detect and stop hate speech on its platform [42]. This study recognizes this void in the existing literature: the predominant focus on the context of COVID-19 and the negligence of the importance of disaggregating online hatred messages directed at Asians.

## 3 DATASETS

### 3.1 Data Collection

We collect 2.6 million messages from X (Twitter at the time of the data collection) using its APIs for academic access. The search period is set from August 2019 to July 2020 to include tweets from pre-COVID-19 and post-COVID-19 peak periods. We use search keywords that are related to Asia and 21 sub-ethnic categories based on the U.S. Census Bureau breakdown.[3] We purposely choose generic keywords to avoid collecting tweets that are only specific to an event (e.g., COVID-19). A complete list of the chosen search keywords is shown in Appendix B.1. With the specified period and keywords, the initial data set includes 10 million tweets, out of which 96.3% of tweets contain with eight major keywords, 'China'(+'Chinese') (31.5%), 'India'(+'Indian') (19%), 'Japan' (+'Japanese') (16.7%), 'Korea'(+'Korean') (11.'%), 'Asia'+('Asian') (10.8%), 'Pakistan'+('Pakistanis') (3.1%), 'Vietnam'+('Vietnamese') (2.3%), and 'Indonesia'+('Indonesian') (1.7%). Other search keywords result in less than 1~2% of the collected tweets.

---

[3]https://www.census.gov/library/stories/2022/05/aanhpi-population-diverse-geographically-dispersed.html

## Table 1: Examples of tweets with high toxicity score but not being toxic towards the search keywords: Tweets that include Asian-related keywords, but do not target them

| |
| --- |
| 1. "We're 1/4 of **China**'s population and we're number 1 in COVID-19 cases, god this country is so fucking shitty" (Score=0.92) |
| 2. "Every fucking human country in world, **CHINA**, **JAPAN**, ENGLAND, ETC has video games!!!! ... Its radical white supremacy..." (Score = 0.92) |

## 3.2 Preprocessing

*3.2.1 PERSPECTIVE API.* Among the Perspective's emotional attributes, we refer to the 'toxicity' score for initial examination of our data. Here, the score lies in between [0, 1], with the highest score 1 being the most toxic. Toxicity is defined as "a rude, disrespectful, or unreasonable comment that is likely to make you leave a discussion". Toxicity is known to result in the most reliable score and has been widely used in previous studies [14, 16]. However, solely relying on toxicity score could both include false positive and omit false negative anti-Asian tweets because anti-Asian sentiment is not always expressed in a toxic manner (see Table 1 for example). Accordingly, in addition to the toxicity score, we introduce a manually annotated label, which indicates whether a tweet contains anti-Asian sentiment. We elaborate it in detail in the following.

*3.2.2 Manual coding.* Although the PERSPECTIVE API provides the scores that reflect the likelihood of assessed tweets being toxic in a reliable manner, it is challenging to see whether the toxic expression was being made towards Asian or specific ethnic groups we are interested in. Likewise, it is possible to dismiss anti-Asian tweets that have low toxicity score. To address this issue, we manually annotate subsampled tweets to obtain more target-indicative information. For subsampling, we first divide the collected tweets into weekly batches and sort them based on the corresponding toxicity scores. From each weekly batch, we randomly sample 20 tweets from ten groups which are broken down based on the toxicity scores (=200 tweets per week), resulting in 10400 tweets in total:

- Group 1: 20 tweets with the scores lie in [0, 0.1],
⋮
- Group 10: 20 tweets with the scores lie in [0.9, 1.0].

Then human annotators manually label the tweets on:

> [**ANTI-ASIAN**] Does this tweet contain "anti-Asian" sentiment? (True/False).

This label **ANTI-ASIAN** is to determine if the negative expression was being directed towards Asian.

*Training annotators.* Graduate student annotators are trained with multiple training sessions, during which they are instructed to make step-wise judgements before annotating the focal attribute. (Step 1) they judge whether a tweet is interpretable at all. (Step 2) they judge whether a tweet is an expression of feeling, thought, opinion, attitude or judgement or perspective about something or someone. (Step 3) only if the tweets meet the first two criteria, they judge whether it is a negative sentiment about Asia, Asian

or Asian-signaling object, with satisfactory inter-coder reliability based on Cohen's kappa =0.882 and percent agreement = 95%.

*Results of the annotation.* After removing illegible, meaningless, or double-edged remarks without providing a context (e.g., "@China_Crazy Instagram"), we keep around 10300 annotated tweets. Among them, 34% contain "anti-Asian" sentiment (**Anti-Asian** is True). We refer readers to Appendix B.2 for details of annotators and the annotation results.

### 3.3 Deep Language Models

To label the remaining tweets that have not been manually annotated, we train and employ deep language models for annotating unlabeled data. We test three deep language models, Bert [7], ELEC-TRA [5], and RoBERTa [22], and choose one that performs the best in a 5-fold cross-validation. For all training and validation tasks, the stratified split of training/validation/test sets as 80/10/10 is considered as there exists class imbalance in the manually annotated label. We find that RoBERTa performs the best with the average validation accuracy of 81.95. For training all models, we use the minibatch of size 32, learning rate 0.0001, and dropout rate 0.15. The patience of 20 epochs for early stopping is employed to prevent the overfitting. All the implementation is based on TensorFlow 2 [1].

## 4 ANALYSIS

### 4.1 Data Statistics

Applying the best performed RoBERTa model results in 383,546 tweets satisfying the condition: [**Anti-Asian** = T] (See Table 2). The average toxicity scores of the tweets is 0.299, which is about 2.4 times larger than that of the counterpart. The rest of analyses are based on the use of these machine-labeled anti-Asian tweets.

**Table 2: The number of tweets with the labels annotated via deep language models and average toxicity scores.**

|  | **Anti-Asian** | | |
|---|---|---|---|
|  | True | False | Total |
| Tweet count | 383,546 | 2,250,141 | 2,633,687 |
| Average toxicity score | 0.298 | 0.124 | 0.149 |

We mainly present results related to messages, referencing Asian, Indian, Korean, Vietnamese, Chinese, Japanese, Pakistani, or Indonesian, are presented because messages that attack other ethnic groups are identified minimally or not at all. Table 3 provides more information, including the averaged toxicity scores for these eight ethnicity references. The toxicity score of Asian is the highest, followed by those of Korean, Japanese, Indonesian, Vietnamese, Chinese, Indian, and Pakistani.

Figure 1 shows the weekly changes in the tweet counts and the average toxicity score for the tweets that satisfy the condition, where Week 1 corresponds to the week starting at Aug 1st, 2019. Figure 1a presents the aggregated weekly tweet counts , in which a big surge occurs in Week 32 (March 12–19, 2020) when the Trump Administration declares a nationwide emergency due to COVID-19. Figure 1c presents the cumulative weekly proportion of tweets containing each ethnicity. Comparable to Figure 1a, Figure 1c shows

**Table 3: Per ethnicity, the total number of tweets (# total), the number of tweets satisfying the condition (# cond), proportions of tweets satisfying the condition (i.e., $\frac{\text{\# cond}}{\text{\# total}}$), and averaged toxicity scores.**

|  | Asian | Chinese | Indian | Japanese |
|---|---|---|---|---|
| # total | 219,690 | 1,000,385 | 461,885 | 387,387 |
| # cond | 19,666 | 230,496 | 96,611 | 9,370 |
| Proportion | 8.95 % | 23.04 % | 20.91 % | 2.41 % |
| Avg. Score | 0.4273 | 0.2795 | 0.3012 | 0.3492 |

|  | Korean | Pakistani | Vietnamese | Indonesian |
|---|---|---|---|---|
| # total | 256,341 | 104,027 | 63,873 | 49,795 |
| # cond | 11,767 | 31,390 | 1,389 | 2,370 |
| Proportion | 4.59% | 30.17 % | 2.17 % | 4.75 % |
| Avg. Score | 0.3576 | 0.3182 | 0.3343 | 0.2910 |

1. Hong Kong protest, India's revocation of the special status of Jammu/Kashmir
2. An outbreak of atypical penumonia-like illness in Wuhan
3. Wuhan lockdown due to the 2019 Novel Coronavirus outbreak
4. Nationwide emergency declared by the Trump Administration in the U.S.

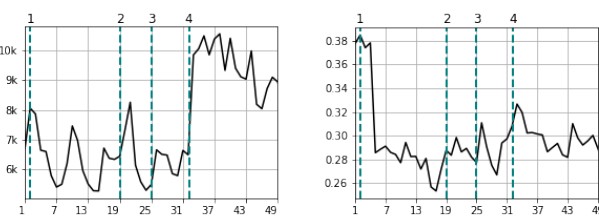

**(a) Tweet counts summed up for the eight major keywords**   **(b) Weekly average toxicity scores for the eight major keywords**

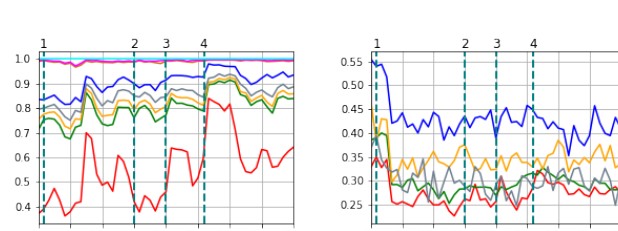

**(c) Proportion of tweets containing each keywords**   **(d) Weekly average toxicity scores of the tweets containing each keywords**

**Figure 1: Weekly counts and average toxicity scores of the tweets with major ethnic keywords that satisfy all conditions**

a peak in the proportion of the Chinese-related tweets in Week 32. However, the aggregated tweet counts in other weeks (Figure 1a) do not necessarily correspond to the peaks of Chinese-related tweets in Figure 1c (e.g., peaks in Week 2 and 22, weeks after Week 32).

Figures 1b and 1d present the weekly average toxicity score, aggregated (Figure 1b) and disaggregated by ethnicity (Figure 1d). We again observe increases in the average scores of overall and China-related tweets in Week 32. However, Japanese and Korean tend to have higher toxicity scores than Chinese over the entire period

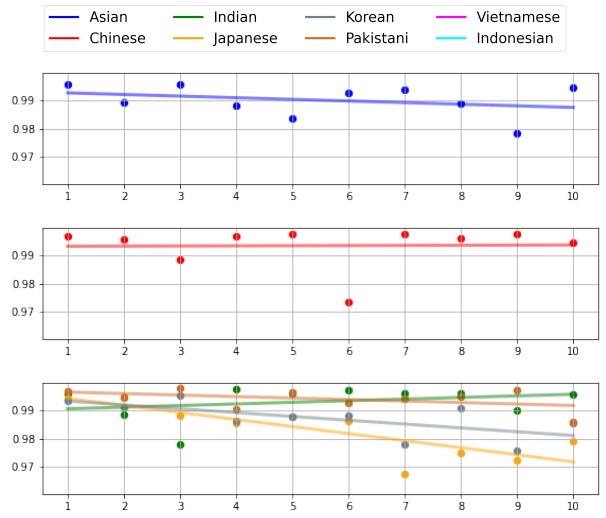

**(a) Asian**      **(b) Chinese**

**(c) Indian**      **(d) Korean**

| — 0-0.1 | — 0.2-0.3 | — 0.4-0.5 | — 0.6-0.7 | — 0.8-0.9 |
| — 0.1-0.2 | — 0.3-0.4 | — 0.5-0.6 | — 0.7-0.8 | — 0.9-1.0 |

**Figure 2: Monthly distribution of toxicity scores towards selected ethnic groups, [Asian, Chinese, Indian, Korean].**

and aligned more with the overall average. More importantly, the figures reveal that the average toxicity was at its highest not during the pandemic but in August 2019, a period when both the protests in Hong Kong were on-going and India revoked the special status of Jammu and Kashmir. While we do not include Vietnamese and Indonesian in the graph to enhance interpretability, see Appendix for Figure1d that includes the two.

## 4.2 Temporal Changes of Toxicity Scores: Persistence Analysis

To explore **RQ1a** about the overtime trends of anti-Asian messages, we investigate the temporal evolution of toxicity scores. For this analysis, we disaggregate tweets by ethnicity using ethnicity-related keywords (See Appendix C.2), construct monthly histograms based on toxicity scores distributions per ethnic group, and perform a statistical analysis to determine the significance in toxicity distribution changes over monthly histograms. Each histogram contains 10 bins with a uniform width, 0.1, i.e., $S^{e,m} = \left[ s^{e,m}_{[0,0.1]}, \ldots, s^{e,m}_{[0.9,1]} \right]$ for an ethnicity group $e$ in month $m$. Here, $s_{[a,b]}$ is the percentage of tweets with toxicity scores ranging from $a$ to $b$.

Figure 2 presents the monthly histograms of toxicity scores towards selected ethnic groups. Consistent with the information in the earlier section, higher toxicity bins take a larger part of the histograms among the Asia group compared to other groups and lower toxicity bins take a larger part of the histograms among the Chinese group.

To examine **RQ1b** about the (dis)similarity of temporal trends across ethnic groups, we use the monthly histograms to statistically measure the consistency in the toxicity scores over time, calculating *persistence scores* [46]. The persistence analysis has been frequently used to capture changes over time, such as dynamical patterns in

**Figure 3: Persistence scores of monthly distribution of toxicity scores.**

spending and consumption of bank customers [41] and emotional changes in Twitter [46]. In our study, we define persistence as the cosine similarity between an ethnicity group's histograms in two consecutive months, i.e., $S^{e,m}$ and $S^{e,m-1}$:

$$P^{e,m} = \text{sim}_{\cos}(S^{e,m}, S^{e,m-1}). \tag{1}$$

Persistence scores range from 0 to 1; the score 1 indicates the highest persistence, meaning that there is no change in the toxicity score distribution between two consecutive months whereas the score 0 indicates the drastic changes. Figure 3 shows the monthly persistence scores (circles) and the fitted line (solid lines) using a linear regression for each ethnicity group.

Several points are worth noting. First, all of the monthly persistence scores are over 0.96, indicating that the distribution of toxicity scores towards each ethnic group is relatively consistent over time. Second, the patterns of toxicity scores are quite different among various groups. In terms of statistical significance, only Japanese and Korean among the eight ethnic groups we observe present a downward trend ($b = -0.0025$, $p = 0.004$; $b = -0.0014$, $p = 0.038$, respectively) although the coefficients are close to zero, suggesting a minuscule change. Also, the results suggest that the change in the distribution of toxicity scores seems to be influenced by events of which the impact are limited to the focal ethnic group. For example, the persistence score for Chinese-referencing messages drops between Month 6 (February 2020) and Month 7 (March 2020 when the COVID-19 was spread all over the world), while it becomes relatively stable at the previous level afterwards. This result, along with the toxicity distribution in the anti-Chinese messages as seen in Figure 2(b), indicates that the distribution of toxicity against Chinese has increased during the peak of COVID-19 and then continued the elevated level afterwards (Figure 3). By comparison, no such trend is shown among other ethnic groups, implying that the COVID-19, or any other events that may have increased anti-Chinese toxic

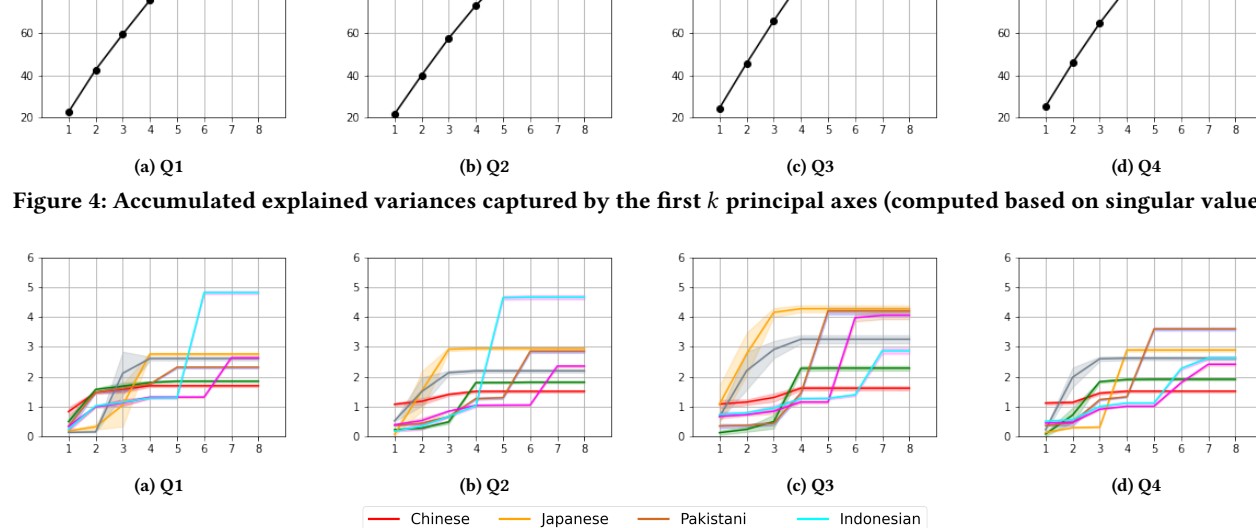

**Figure 4: Accumulated explained variances captured by the first $k$ principal axes (computed based on singular values).**

**Figure 5: Mean and standard deviation of distance to Asia measured in the embedding spaces with increasing dimensions (i.e., incrementally adding principal axes up to the 8-dimensional space).**

messages during early 2020, have not influenced the distribution of the toxic messages that target other ethnic groups.

In sum, we find that while the toxicity of Asian-referencing messages is largely stable over time, nuanced differences exist in the temporal patterns when the data are disaggregated by ethnicity.

## 4.3 Semantic Distances among anti-Asian Messages: Multiple Correspondence Analysis

**RQ2s** examine semantic distances among anti-Asian messages that target different ethnic groups (**RQ2a**) and how these distances vary over time (**RQ2b**). To address RQ2s, we break down the dataset into quarterly datasets. We perform multiple correspondence analysis (MCA) [18] on these quarterly datasets. MCA reveals an underlying structure or relationships of nominal categorical variables; in short, the closer the variables are in the $d$-dimensional Euclidean space, the more semantically similar they are. Based on the results of MCA, we investigate cross-ethnic differences in terms of the distances from the [Asian] variable to the other ethnicity categorical variables in the embedding space.

To perform MCA, we first construct a contingency table whose columns consist of the major ethnic group variables [Chinese, Indian, Japanese, Korean, Asian, Pakistani, Vietnamese, Indonesian], and whose rows consist of the $n$-grams (uni-, bi-, and tri-grams) that appear in the dataset. Once the contingency table is constructed, a singular value decomposition is applied to the preprocessed matrix to obtain orthogonal vectors that represent the ethnic group variables. For this analysis, we focus on explicitly toxic tweets by setting the toxicity score threshold to be $\tau = 0.8$. We repeatedly apply MCA to quarterly datasets, Q1, Q2, Q3, and Q4, with varying hyper-parameters ($n$ in $n$-grams, etc) and report statistical quantities

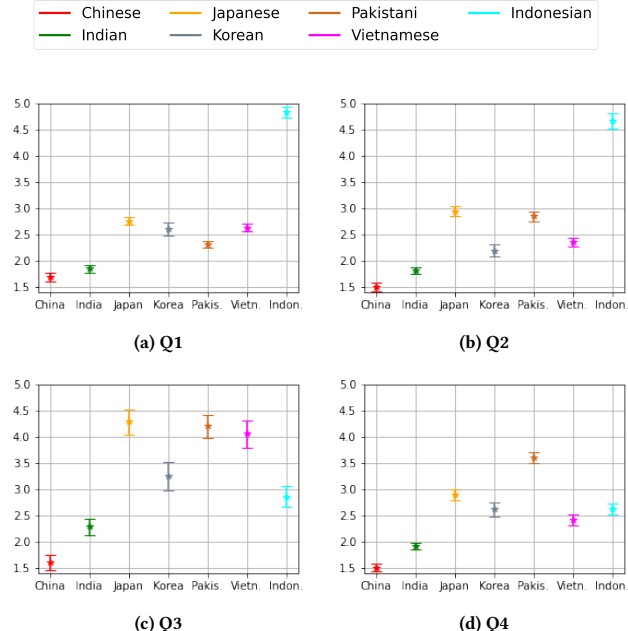

**Figure 6: Mean and standard deviation of distance to Asia in the 8-dimensional embedding space, the space spanned by all 8 principal axes.**

(mean and standard deviation) of results. We refer readers to Appendix for more details on preprocessing (C.4.1), hyper-parameters (C.4.2) and additional results (C.4.5).

Figures 4–6 show the results of the MCA. Figure 4 shows the accumulated explained variances captured by increasing the number of principal axes; the principal axes are sorted in a decreasing order based on the explained variance that each principal axis captures; that is, the largest explained variance is captured by the first principal axis. Figure 4 essentially shows that ∼90% and ∼99% are captured by the first five and six principal axes for all quarters. Figure 5 shows the distances from the categorical variable [Asian] to other categorical variables [Chinese, Indian, Japanese, Korean, Pakistani, Vietnamese, Indonesian], measured in the Euclidean distance (i.e., the L2-distance) in the embedding space generated by MCA. We vary the dimensionality of the embedding space from $d = 1$ to $d = 8$ and Figure 6 summarizes the distances from [Asian] to other variables with $d = 8$ (which is the full space as there are 8 categorical variables).

We make some notable observations from Figures 4 and 5. First, for all quarters, the distances to three categorical variables [Pakistani, Vietnamese, Indonesian] increase by adding the 5, 6, and 7th principal components, while the distances to other ethnicities [Chinese, Indian, Japanese, Korean] remain unchanged after the 4th principal component. This observation suggests that there are some discussions relevant to [Pakistani, Vietnamese, Indonesian] that are orthogonal to other ethnicities, which makes the distance to these categorical variables greater. Second, [Indian, Pakistani] and [Japanese, Korean], respectively, tend to be similarly affected by the same principal components, suggesting that there are some discussions that are common between Indian and Pakistani, and Japanese and Korean, respectively ([Japanese, Korean] in Q4 appears to be an exception, though). Third, the categorical variable [Chinese] has the closest distance to [Asian] in all quarters (Figure 6). This finding appears to be partly driven by the fact that anti-Chinese toxic messages take the largest volume in the sample, which makes the its distance to [Asian] the closest among other messages that target other ethnicities. Moreover, in Q3, Asian's distances to other ethnicities (i.e., except [Chinese]) tend to become larger than those in other quarters (Figure 6c), which suggests that the influence of the COVID-19 during the peak pandemic phase is concentrated to targeting the Chinese ethnicity. This makes China-hate messages semantically more distant from messages attacking other Asian subethnicities. Fourth, while [Chinese] has the closest distance to [Asian], its distance is not the closest when including only the first 4 or 5 principal components (explaining 80% or higher variances). This finding suggests that frequent topics of anti-China messages are different from those of anti-hate discussions of other ethnicities.

In sum, the results from the MCA suggest that toxic anti-Asian messages encompass a range of discussions that vary over time and depending on the targeted sub-Asian ethnicities.

## 4.4 Topic Similarities among Anti-Asian Messages: BERTᴏᴘɪᴄ Modeling

To further investigate topical narratives in anti-Asian tweets (RQs 3), we perform topic modeling. Topic modeling supplements the $n$-gram based assessment in the earlier MCA section by enabling the examination of actual narratives of anti-Asian messages. To perform topic modeling, we use the BERTᴏᴘɪᴄ API [15], a Transformer-based topic modeling technique that provides human-interpretable results.

We choose BERTᴏᴘɪᴄ over other alternatives such as latent Dirichlet allocation (LDA) or non-negative matrix factorization (NMF) because BERTᴏᴘɪᴄ outperforms the other methods (LDA, NMF) in terms of two performance measures, topic coherence and topic diversity (see Appendix C.5 for the definitions and performance outcomes of these metrics).

BERTᴏᴘɪᴄ takes a collection of documents, embeds the documents into vector representations, reduces them via dimensionality reduction to cluster them, and computes latent topics via identifying the most representative words in each cluster. In our modeling, we consider Sentence-Transformer [32], UMAP [25], and HDB-SCAN [24] for document embedding, dimensionality reduction, and clustering. We run BERTᴏᴘɪᴄ model instances with 100 combinations of various hyper-parameter settings. We report the results in statistics. For descriptions on preprocessing and the considered hyper-parameters, we refer readers to Appendix.

To be consistent, we apply the same threshold in data selection as in the MCA (i.e., the toxicity scores greater than or equal to 0.8). Given that the total volume of tweets exceeding the toxicity score of 0.8 is not substantial, we group the sample into four categories for topic modeling: [Asian, Chinese, Indian, and Other Asian (i.e., the union of Japanese, Korean, Pakistani, Vietnamese, and Indonesian)] and without temporal partitioning. The Chinese and Indian groups are compared separately due to their relatively large message volumes.

Topic modeling results in the probability score of each topic within each tweet, which describes how likely a tweet contains a given topic. As a total of 30 topics are inferred from the topic modeling, 30-dimensional vector is given to a tweet, where an element of the vector describes a probability of the tweet being assigned to a topic. After assigning topic probabilities within each tweet, we disaggregate the dataset by splitting tweets into four ethnicity-based groups [Asian, Chinese, Indian, OtherAsian], based on the same keyword-based selection process, as described in the earlier section (and also detailed in Appendix C.2). Finally, we average the topic probabilities (the 30-dimensional vector) assigned to each tweet in a group-wise manner, resulting in four averaged topic probabilities associated with each group.

First, before examining the contents of topics, we perform statistical tests using the Spearman's rank-order correlation coefficients to measure topical similarity between messages that broadly target Asian in general and those that target other groups, [Chinese, Indian, OtherAsian], respectively. The higher the coefficient is, the more similar the rank order of topic probabilities between the two compared groups is. The results suggest that messages broadly targeting Asian in general ([Asian]) have a more similar topic rank-order to that of the OtherAsian group ($\rho$ =0.688, $p$ =0.004), i.e., the collection of messages directed at relatively small-sized ethnic groups rather than to that of the large ethnic groups, Chinese ($\rho$ =0.398, $p$ =0.033) and Indian ($\rho$ =0.430, $p$ =0.020). This observation suggests that [OtherAsian] has the closest topical distance to [Asian]. This point is also consistent with the fourth finding in the MCA; that is, [China] or [India] are not the closest group to [Asian] in the low-dimensional space (i.e., $d \leq 4$) where the principal axes are relevant to narratives that are common to all groups.

Next, we examine the topics that yield the highest average probability within each group. Table 4 presents the most representative

**Table 4: Representative topics obtained from BERTopic (the obvious words, e.g., 'China' in the Chinese group's topic, are omitted from the representative words).**

| Group | Topic | Prob. | Representative words |
|---|---|---|---|
| Asian | Asian-on-other-race-trop | 0.474 | black, white, racist |
| Chinese | Hate against Chinese community | 0.408 | realdonaldtrump, communist, government |
| Indian | India-Pakistan tension | 0.643 | terrorist, muslim, country |
| OtherAsian | Blasphemy surrounding K-pop | 0.228 | kpop, fan, bitch |
|  | Anti-Pakistan | 0.187 | terrorist, muslim, country |

topics of each group with their topic probability. See Appendix C.5 for top-5 topics in each group with example tweets. First, we find that the most predominant topics for each group are different from one another. The most frequently discussed anti-Asian narratives are uniquely shaped by 'whom' the message attacks.

Among the messages that broadly target Asian in general ([Asian]), the topic with the highest average probability score contains themes related to domestic inter-racial conflicts. By comparison, the most likely topics directed at Chinese and Indian revolve around global politics and ideological tensions, including expressions of anti-communism and Hindu-Muslim conflict, respectively. In all of these three groups, each of the most prominent topic stands out with a substantially higher probability score than the topic with the second-highest probability score (e.g., the highest and the second high scores of topics in the Asian group are 0.474 and 0.090). On the other hand, within the OtherAsian group, the topic probabilities are distributed more evenly. The topic that earns the highest score was negative attitudes towards K-pop culture with the score of 0.228, followed by anti-Pakistan narratives, with the score of 0.187, showing only a 0.04 percentage point difference between them. These results suggest that a unique topic highly dominates hate narratives towards the Chinese and the Indian group, respectively, which makes their topical distance farther away from those topic narratives directed at Asian in general, as evidenced in Table 4.

In sum, findings from the topic modeling suggest that there are distinct and pronounced thematic differences in the narratives targeting different groups, with varying degrees of intensity and focus on specific topics. Understanding these variations is essential for grasping the diversity of perspectives and concerns within the larger Asian community.

## 5 DISCUSSION AND CONCLUSION

This study takes a disaggregated data practice approach to examine online anti-Asian hate, in line with the emphasis that policymakers have placed on gaining a more comprehensive understanding of Asian communities. Drawn from three analytic techniques– toxicity score-based persistence analysis, $n$-gram based MCA, and topic modeling-based Spearman's rank correlation– help deepen our understanding of anti-Asian hate that occurs online. We disaggregate the dataset based on the two axes of temporality and ethnicity, which allow us to identify specific patterns in the changes in toxicity levels of anti-Asian messages directed at various sub-ethnic groups. Moreover, the identification of unique orthogonal clusters of hate messages targeting minority Asian ethnic groups, as revealed by

the MCA results as well as evidenced by the topic analysis, reiterates the importance of data disaggregation. Overall, the findings highlight the distinct nature of anti-Asian hate directed at various ethnic groups, reaffirming the need for a nuanced computational approach in addressing the issue of anti-Asian hate.

Our approach of using various methodological techniques requires careful consideration as different analytical techniques may yield varying insights when assessing the problem of anti-Asian hate. For example, the $n$-gram-based MCA with granular data disaggregation suggests that hate messages targeting larger ethnic groups, such as Chinese and Indian, are semantically close to those targeting Asian in general, when all of the eight principal components are included even though they are not as close when only the first 4 or 5 principal components. This result may have been influenced by the sheer volume of anti-messages targeting the larger groups. By comparison, the application of rank-order correlation tests using topic modeling outputs is less sensitive to the relative data size and suggests that prominent narratives in messages targeting smaller ethnic groups are more similar to the narratives of hate messages targeting Asian in general, as opposed to those specifically targeting Chinese or Indian communities. As such, it is important to consider appropriate techniques and models that align with specific objectives and interests to identify patterns of data for effective data disaggregation practices. For example, if one should weigh the absolute volume of conversations in their assessment, $n$-gram based MCA would be a more appropriate technique than topic modeling-based Spearman's rank correlation. Conversely, if the focus is on emphasizing the actual discursive content, topic modeling and Spearmans' rank correlation may provide more nuanced insights than $n$-gram based MCA.

Regarding the data size imbalance across ethnicities, it is also worth to note that a limitation lies in the nature of historical data collection as opposed to real-time data collection. The platform may have already filtered out some of highly toxic tweets before our data collection, and its moderation could have served majority ethnicities better than minority ethnicities.

Having said that, one of the significant takeaways from this study is the broader applicability of disaggregated data practices. While this study primarily focuses on anti-Asian hate, "panethnic" communities are prevalent globally, encompassing various subset of world populations. The universal applicability of disaggregated data practices in addressing social issues relevant to panethnic communities is a noteworthy aspect. It emphasizes the broader significance of this research beyond the specific context of anti-Asian hate.

In conclusion, this study has highlighted the importance of disaggregating data to gain a more nuanced understanding of online anti-Asian hate. The findings underscore the complexities and unique challenges faced by marginalized Asian communities. By scrutinizing nuanced ethnicity-based hatred, this study encourages critical reflection on inter-ethnic relations and corresponds to a multicultural society's needs to value diversity, equity, and inclusion.

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

## A  BROADER PERSPECTIVE, ETHICS AND COMPETING INTERESTS

By scrutinizing nuanced ethnicity-based hatred, this study encourages critical reflection on inter-ethnic relations and corresponds to a multicultural society's needs to value diversity, equity, and inclusion. While it is important to look into the nature of hate speech, we also acknowledge a possibility to cause unintended priming effects by surfacing the details of undesirable messages. This concern may apply not only to the current study but to all public research and media coverage that report incidents of hate and toxic messages.

We collected data complying with the protocol approved by the Institutional Review Boards (IRBs) at the researchers' institutions that ensures user privacy; we have collected data complying with the protocol that ensures user privacy; 1) Twitter master ID-list is separately restored and 2) only tweets and their timestamps have been used for analysis, i.e., user profiles have not been utilized for analysis. We plan to release this dataset publicly available with the stipulation that those who use it must comply with X's Terms and Conditions and must not attempt to (de)-identify user profiles.

## B  MORE DETAILS ON DATASET COLLECTION

### B.1  Search Keywords

Table 5 lists a complete set of search keywords. We use search keywords that are related to Asia and 21 sub-ethnic categories based on the U.S. Census Bureau breakdown[4].

**Table 5: Twitter search keywords (alphabetically-ordered)**

| Ethnicity-based search keywords |
|---|
| Asia, Asian, Cambodia, China, Chinese, Filipino, Hmong, India, Indian, Indonesia, Indonesian, Japan, Japanese, Korea, Korean, Laos, Laotian, Malaysia, Malaysian, Mongol, Mongolian, Okinawan, Nepal, Nepalese, Pakistan, Pakistani, Philippine, Sri Lanka, Sri Lankan, Thailand, Vietnam, Vietnamese |

We also attempt to collect tweets containing Asian-targeting slurs for which we reference the WIKIPEDIA article[5]; the keywords used include ['abcd','banana','buddhahead','charlie','chinaman','ching chong', 'chink', 'coconut', 'coolie', 'dink', 'flip', 'gook', 'gook-eye', 'gooky', 'hajji', 'hadji', 'haji', 'jap', 'nip', 'slope', 'slopehead', 'slopy', 'slopey', 'sloper', 'slant', 'slant–eye', 'twinkie', 'zip', 'zipperhead']. However, no tweets including such keyword are collected except the ones containing the general meanings such as 'coconut'. We suspect that tweets including such words have already been removed from the archive as they do not appear in our search. This can be considered as a limitation regarding the use of a keyword-based sampling, which we further elaborate in the following.

*Limitation on a keyword-based sampling.* Even if a keyword-based sampling is widely used and often an essential step for text mining in social media, there is an unavoidable constraint due to an "undocumented upper limit known as streaming cap" [9], however

a researcher builds an extensive keyword list. Further, a static set of keywords may not capture evolution of language uses such as appearances of new words or (sometimes intentional) misspellings. Although we may lose some information that can be obtained from those non-permanent terms, we choose to include only general and permanent terms to reliably perform longitudinal analysis.

### B.2  Annotation result details and potential limitation

Two doctoral students (one male and one female, Chinese descendants) in journalism/communication were annotators, with satisfactory intercoder reliability: Cohen's Kappa = 0.882, percent agreement = 95% for **Anti-Asian**, respectively. A random subset of manually coded tweets were further reviewed for validation by the authors–a mixture of genders, ethnicities (Indian, Korean, and Chinese), and age (20s-40s).

Although we strived to provide a reliable and generalizable dataset, online hate is essentially a nuanced and subjective construct and annotators' experiences could have influenced the annotation output.

## C  DETAILS ON ANALYSIS TOOLS

### C.1  Perspective API

Perspective is an API developed by Jigsaw[6] and Google's Counter Abuse Technology team under a collaborative research initiative called Conversation-AI. Perspective API scores the perceived impact a comment (e.g., a tweet on Twitter) might have on a conversation by using machine learning models. The perceived impact is evaluated by assessing a variety of emotional concepts, denoted as *attributes*, including toxic, insulting, threatening, and so on. The score on each attribute is represented as a numerical value between 0 and 1, representing a probability; the higher the score, the greater the likelihood that a reader would perceive the comment as containing the given attribute. The machine learning models are trained with the probability scores that have been manually coded by the crowdsourced human annotators. To be more precise, the probability scores are marked as the ratio of raters who tagged a comment as the one that contains one of the attributes; for example, if 6 out of 10 annotators tagged a comment as toxic, 0.6 is given to the comment as its probability score.

Figure 7 depicts the weekly average toxicity scores of the tweets with all ethnic keywords that satisfy all conditions.

### C.2  Ethnicity Grouping based-on Keywords

Ethnicity-specific groups are defined based-on ethnicity-related keywords. Each group is mutually exclusive, meaning that for constructing each dataset, tweets containing the following keywords exclusively are collected:

- Asian: "Asia", "Asian", "Asian's",
- Chinese: "China", "Chinese", "China's",
- Indian: "India", "Indian", "India's",
- Japanese: "Japan", "Japanese", "Japan's",
- Korean: "Korea", "Korean", "Korea's",
- Pakistani: "Pakistan", "Pakistanis", "Pakistan's",

---

[4]https://www.census.gov/library/stories/2022/05/aanhpi-population-diverse-geographically-dispersed.html
[5]https://en.wikipedia.org/wiki/List_of_ethnic_slurs

[6]https://jigsaw.google.com/

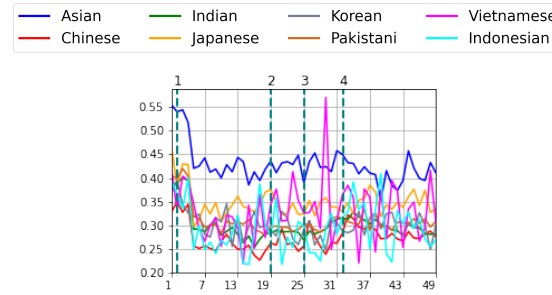

**Figure 7: Weekly average toxicity scores of the tweets with major ethnic keywords that satisfy all conditions**

- Vietnamese: "Vietnam", "Vietnamese", "Vietnam's",
- Indonesian: "Indonesia", "Indonesian", "Indonesia's".

For example, the "Chinese" group includes tweets containing the keywords, "China", "Chinese", "China's", but not other ethnicity-related keywords.

For computational analysis, we further downsample the groups based on the tweets' toxicity scores. We use three values $\tau = \{0.7, 0.8, 0.9\}$ for thresholding the groups and keep only the tweets that satisfying the condition, the toxicity score $\geq \tau$. Figure 8 shows the per-ethnicity counts and averaged toxicity scores of tweets filtered based on the toxicity score threshold $\tau = \{0.7, 0.8, 0.9\}$.

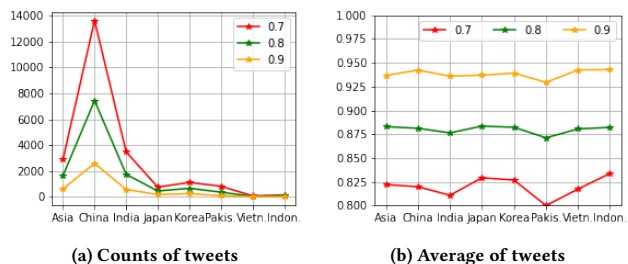

| (a) Counts of tweets | (b) Average of tweets |
|---|---|

**Figure 8: Counts and average of tweets with the toxicity score greater than or equal to a threshold $\tau = \{0.7, 0.8, 0.9\}$.**

## C.3 Persistence Analysis

## C.4 MCA

MCA is a statistical technique to reveal the underlying structure or the relationship of nominal categorical data; MCA operates similarly with the principal component analysis (PCA) for continuous-values data, representing the data as points in a low-dimensional Euclidean space identified by a set of important vectors. In short, the closer the variables are in the low-dimensional Euclidean space, the more semantically similar they are.

To perform MCA, we first construct a contingency table, whose columns consist of the major keywords, [China, India, Japan, Korea, Asia, Pakistan, Vietnam, Indonesia], and whose rows consist of the $n$-grams (uni-, bi-, and tri-grams) that appear in the tweets containing each major keyword. Once the contingency table is constructed,

standard preprocessing (including centering) steps to the contingency table is followed and, finally, a singular value decomposition is applied to the resulting matrix to obtain orthogonal vectors that represent the categorical variables (such as in PCA).

*C.4.1 Text preprocessing for n-grams.* To compute $n$-grams, we first apply following preprocessing to clean up texts: (1) url and HTML tags are removed, (2) the texts are lower cased and special characters along with unnecessary tabs and white spaces are removed. (3) emojis are removed, (4) decontraction of the text is performed (e.g., from "I've" to "I have"), and (5) finally, English stopwords defined by Natural Language Toolkit (NLTK) [2] are removed.

*C.4.2 Hyper-parameters.* The hyper-parameters we consider in MCA are:

- $n\_gram$ in $\{1, 2, 3\}$, which represents three combinations of $n\_gram$, '1' denotes uni-gram, '2' denotes the combination of uni-gram and bi-grams, and '3' denotes the combination of uni-gram, bi-grams, and tri-grams.
- $cutoff$ in $\{5, 10, 15, 20\}$, which represents the minimum total frequency of words across different ethnicities. The frequency of words below the cutoff values are considered as insignificant to analyze.

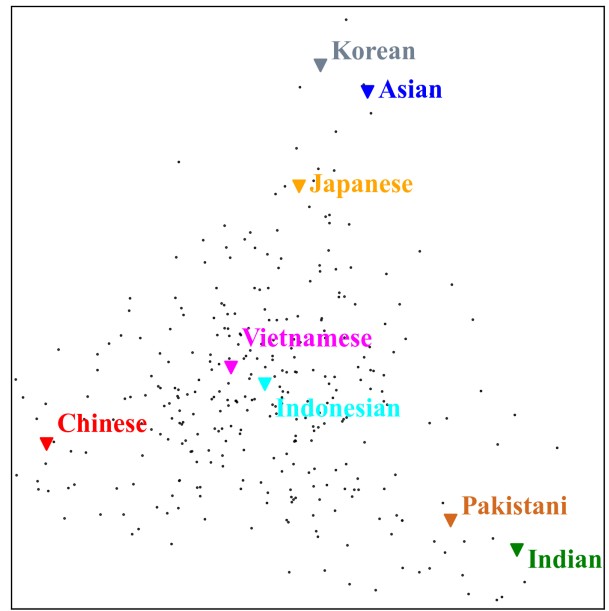

**Figure 9: [MCA] 2-dimensional representation of the categorical variables.**

*C.4.3 A visual example of MCA results.* Figure 9 shows an example result of applying MCA to the contingency table: The ethnicity variables and the $n$-grams are projected in the two-dimensional embedding space, a space spanned by the first two principal axes. Here, the principal axes are sorted in a decreasing order based on the explained variance that each principal axes captures; that is, the largest explained variance is captured by the first principal axes. The distances presented in Figure 5 are measured in the Euclidean

**Table 6: Ranks determined in distances from Asian to other ethnicities in the 8-dimensional embedding space (the closest one has the rank one).**

| Rank | Q1 | Q2 | Q3 | Q4 |
|---|---|---|---|---|
| 1 | Chinese | Chinese | Chinese | Chinese |
| 2 | Indian | Indian | Indian | Indian |
| 3 | Pakistani | Korean | Indonesian | Vietnamese |
| 4 | Korean | Vietnamese | Korean | Korean |
| 5 | Vietnamese | Pakistani | Vietnamese | Indonesian |
| 6 | Japanese | Japanese | Pakistani | Japanese |
| 7 | Indonesian | Indonesian | Japanese | Pakistani |

distance in each $d$-dimensional spaces, spanned by the first $d$ principal axes (i.e., the number of the principal axes are incrementally added when $d$ gets increased).

*C.4.4   Extra information on the result with $\tau = 0.8$.* Table 6 presents the orders of the ethnicities in each quarter sorted by the distance from the Asian variable; the distance is measured in the full 8-dimensional space. Table 6 shows that the ranks of the two closest ethnicity variables, 'Chinese' and 'Indian', are invariant over the period of the investigation, while those of other variables are varying.

*C.4.5   Additional Results.* Figures 10 and 11 show the distances from the categorical variable [Asia] to other categorical variables [China, India, Japan, Korea, Pakistan, Vietnam, Indonesia], measured in the Euclidean distance (i.e., the L2-distance) in the embedding space generated by MCA. To filter the tweets for each ethnic group, different values of toxicity score threshold are employed in these experiments, i.e., $\tau = 0.7, 0.9$. Figures 12 ($\tau = 0.7$) and 13 ($\tau = 0.9$) show the accumulated explained variances captured by increasing the number of principal axes; again, ~90% and ~99% are captured by the first five and six principal axes for all quarters.

## C.5   Topic modeling

We evaluate the model performance by utilizing two commonly used metrics, *topic coherence* (TC) and *topic diversity* (TD) that operate on the top 10 words of top 10 topics. After training the topic models (BERTopic, LDA, NMF), a topic is represented by $n$ words that have the highest probability of association with that specific topic. TC measures the interpretability of topics for human comprehension; a greater resemblance among the words within a topic corresponds to a higher coherence. The evaluation of TC for the topic model is conducted using the normalized pointwise mutual information (NPMI) [3], a metric ranging from -1 to 1, where -1 implies that the top $n$ words never occur together within a topic, 0 denotes independence, and 1 indicates that the top $n$ words are completely co-occurrence. TD assesses the distinctiveness of topics, quantified by the percentage of unique words of top 10 words in top 10 topics [8]. TD ranges in [0,1], where 0 indicates redundant topics and 1 indicates more various topics. A higher topic diversity implies better coverage of various aspects within the analyzed corpus.

Table 7 demonstrates that BERTopic outperforms the other two models, achieving the highest scores for both TC and TD. Table 8

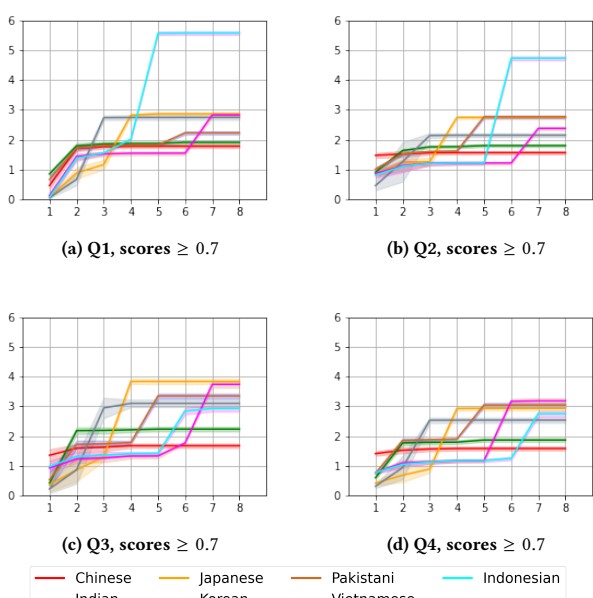

(a) Q1, scores $\geq 0.7$          (b) Q2, scores $\geq 0.7$

(c) Q3, scores $\geq 0.7$          (d) Q4, scores $\geq 0.7$

Chinese    Japanese    Pakistani    Indonesian
Indian     Korean      Vietnamese

**Figure 10: Mean and standard deviation of distance to Asia measured in the embedding spaces with increasing dimensions (i.e., incrementally adding principal axes up to the 8-dimensional space).**

further investigates the performance of three different topic modeling approaches with a value for thresholding the toxicity score $\tau = \{0.7, 0.8, 0.9\}$; the table essentially shows that BERTopic produces the best results in terms of TC and TD. We note that in all three methods, topic diversity becomes worse with $\tau = 0.9$ as the number of remaining tweets becomes decreased.

**Table 7: Topic coherence and topic diversity**

| | TC | TD |
|---|---|---|
| BERTopic | 0.1562 | 0.92 |
| LDA | 0.0176 | 0.73 |
| NMF | 0.0313 | 0.59 |

**Table 8: Topic coherence and topic diversity of three different topic modeling approaches**

| | BERTopic | | LDA | | NMF | |
|---|---|---|---|---|---|---|
| $\tau$ | TC | TD | TC | TD | TC | TD |
| 0.7 | 0.0725 | 0.82 | -0.0055 | 0.47 | 0.0129 | 0.52 |
| 0.8 | 0.0747 | 0.89 | -0.0344 | 0.37 | -0.0006 | 0.53 |
| 0.9 | 0.0152 | 0.66 | -0.0378 | 0.32 | -0.029 | 0.46 |

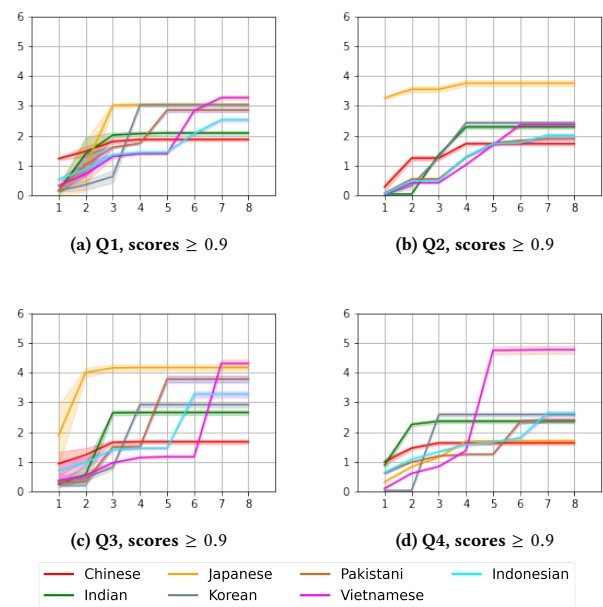

(a) Q1, scores $\geq 0.9$                 (b) Q2, scores $\geq 0.9$

(c) Q3, scores $\geq 0.9$                 (d) Q4, scores $\geq 0.9$

| Chinese | Japanese | Pakistani | Indonesian |
| Indian | Korean | Vietnamese | |

**Figure 11: Mean and standard deviation of distance to Asia measured in the embedding spaces with increasing dimensions (i.e., incrementally adding principal axes up to the 8-dimensional space).**

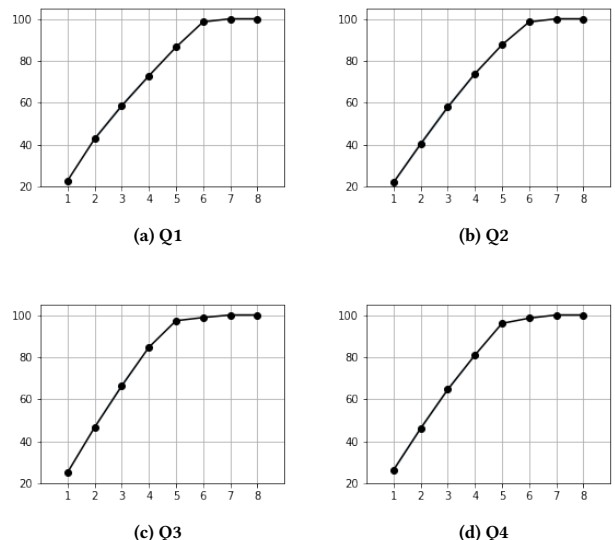

(a) Q1                 (b) Q2

(c) Q3                 (d) Q4

**Figure 12: Accumulated explained variances captured by the first $k$ principal axes (computed based on singular values).**

*Preprocessing.* Each tweet is preprocessed by using the OCTISAPI[7] to lemmatize and remove stop words.

[7]https://github.com/MIND-Lab/OCTIS/tree/master

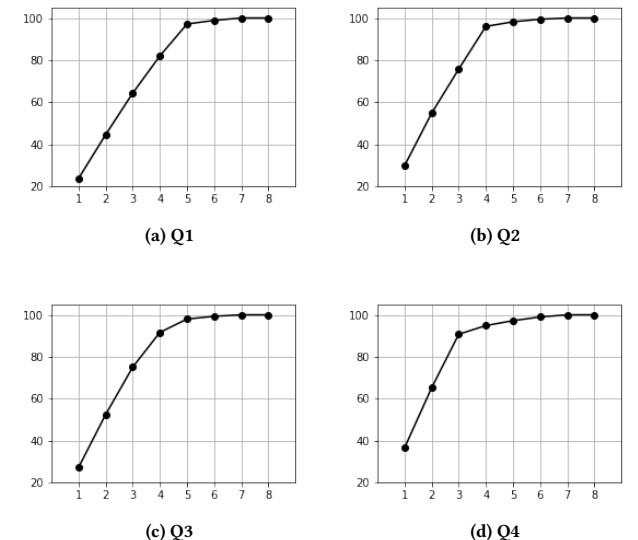

(a) Q1                 (b) Q2

(c) Q3                 (d) Q4

**Figure 13: Accumulated explained variances captured by the first $k$ principal axes (computed based on singular values).**

*Hyper-parameters.* The following list describes the hyper-parameters and their meanings:

- *n_components* is a parameter of UMAP that determines the dimensionality of embedding that is reduced into.
- *n_neighbors* is a parameter that determines the size of the local neighbors that UMAP will look at to learn the manifold structure of the data. A high value of *n_*neighbors will force UMAP to consider a global view, which may lose some detailed information. Conversely, a low value of *n_*neighbors makes UMAP focus on a very small-scale structure.
- *min_cluster_size* is a parameter of HDBSCAN that determines the smallest group size that is considered to be a cluster. A larger *min_cluster_size* will reduce the number of clusters by merging some clusters together.
- *min_samples* is a parameter of HDBSCAN that measures the conservative of the cluster. The larger *min_samples*, the more conservative the cluster, which means that more points will be considered as noise that are not clustering.
- *min_df* is a parameter that indicates the minimum frequency of words when building the vocabulary. If the number of documents in the dataset that have a specific word is lower than the *min_df*, then the word will be ignored. A lower value of *min_df* may contain some words that cannot represent enough information. A higher value of *min_df* may include some words that occur too frequently but are meaningless as the topic representation.
- *nr_topic* is a parameter that indicates the number of topics that will be reduced after training the topic model. After training BERTopic, if the number of topics is higher than *nr_topic*, then the number of topics will be reduced to equal to *nr_topic*. If the number of topics is lower than or equal to *nr_topic*, no reduction will be applied.

For generating the results in Tables 7 and 8, we consider hyper-parameter combinations: $n\_components \in \{5, 10\}$, $n\_neighbors \in \{5, 10, 15, 20, 50\}$, $min\_cluster\_size \in \{10, 20\}$, $min\_samples \in \{1, 10, 20\}$, $min\_df \in \{5, 10\}$, and $nr\_topic \in \{50, 100\}$. For generating the results in the main text, we consider the same hyper-parameter combinations except $nr\_topic \in \{30, 50, 100\}$. With the larger values of $nr\_topic$ (i.e., 50, 100), BERTopic starts to extract less meaningful topics; for examples, topics including only a single tweet.

*Additional results.* Tables 9–12 list the representative topics, corresponding probability, and example tweets in [Asian, Chinese, Indian, OtherAsian] data. As noted in the main text, there exist standing-out (in terms of topic probability) topics in [Asian, Chinese, Indian] while in the OtherAsian data, the topic probabilities are distributed more evenly between different topics.

**Table 9: Representative topics in the "Asian" group**

| Topic | Prob. | Example Tweet |
|---|---|---|
| 1. Asian-on-other-races trope | 0.474 | "this is an attack by racist **asian** scumbags on young white men they need locking up our society is well and truly fucked" |
| 2. General expressions of hate | 0.090 | "lets kill asians" |
| 3. Derogation of food culture | 0.036 | "if i have to see that weird asian bitch eat something alive or gross anymore on the internet i am gonna be the first person ever to defeat the internet" |
| 4. Asian-on-black trope | 0.027 | "need more black beauty stores fuck these racist ass asian people" |
| 5. Virus-related hate | 0.023 | "why the fuck is it always asian countries that come up with these crazy ass sshit diseases lol" |

**Table 10: Representative topics in the "Chinese" group**

| Topic | Prob. | Example Tweet |
|---|---|---|
| 1. Hate against Chinese communism | 0.408 | "donaldjtrumpjr realdonaldtrump joebiden your ridiculous father is still praising china you asshole while people protest and want democracy your daddy is kissing china s ass your sister feeds her family using child labor and begging china for trademarks shut your tiny little mouth" |
| 2. Derogation of food culture | 0.051 | "all of this because of those stupid chinese people who eat anything they see even human stool like they lack food to eat avoid all chinese people before they eat you alive" |
| 3. Virus-related hate | 0.049 | "chuckcallesto hell to the yes i even refuse to call it covid 19 i commonly refer to it as the chinese virus crap chinesevirus" |
| 4. China's dealing with muslims | 0.018 | "oh and it s cute and rather revealing how this maggot motherfucker doesn t care about the muslims china tortures and murders anything to own trump no wonder majid can t keep a fucking job" |
| 5. anti-business that partner with Chinese company | 0.010 | "a big fuck you to blizzard_ent for going against freedom of speech and supporting the oppressing chinese government i really wish your company bankruptcy fuck you and fuck your games i ll never buy a blizzard game ever in my life" |

**Table 11: Representative topics in the "Indian" group**

| Topic | Prob. | Example Tweet |
|---|---|---|
| 1. india-Pakistan tension | 0.643 | "you bunch of liars how come we end up with a bunch of east indian and pakis that treat their women like s and end up pouring gas on them or drowning them or killing him outright" |
| 2. Anti-globalism | 0.034 | "jimmydox2 jhcansouth seanhannity 1 no globalism 2 little kids in chink and indian sweatshops are the reason its so cheap the price jump is people getting paid for labor" |
| 3. Blasphemy due to lagging in game/computing | 0.015 | "fix ur garbage game u stupid indian i keep lagging" |
| 4. Derogation of Indian Tiktok | 0.004 | "this is the same shit as those indian tik toks" |
| 5. Mistreatment of muslims in India | 0.002 | "mkula welcome to malaysia zakir naik in india is bullshit country for you this country is safe for you nobody will harm you even this lol minister in india they were killing muslim for no reason that a muslim cannot eat cow here can eat everyday" |

**Table 12: Representative topics in the "OtherAsian" group**

| Topic | Prob. | Example Tweet |
|---|---|---|
| 1. Blasphemy surrounding K-pop | 0.228 | "tomhollandisoverparty let me guess this is another k fuck oh sorry kpop douche fuck thing jesus fuck stay on korea with you weak ass loser shit this is what our pussified society has become a bend over up the ass society to korean pop fucks great" |
| 2. Anti-Pakistan | 0.187 | "fawadchaudhry seriously this asshole is the minister of science in pakistan no wonder you only produce terrorists" |
| 3. Anti-communism/ authoritarinism | 0.122 | "deplorablereeg1 patrici76267702 i am ready to hunt amp kill communists at any time i did it in vietnam and i would do it again fucking sons of bitches" |
| 4. Anti-Japan | 0.104 | "crunchy roll said fuck the japanese black people made this shit" |
| 5. Asian-on-Black | 0.089 | "brooooo the fucking vietnamese coworker always saying it always the black guy fault and our black coworker is here listening to him like bruh this man is a savage" |

Received 20 February 2007; revised 12 March 2009; accepted 5 June 2009

