# OpenReview forum: "Not All Asians are the Same: A Disaggregated Approach to Identifying Anti-Asian Racism in Social Media"
_ACM.org/TheWebConf/2024/Conference — TheWebConf24_

### Official Review · Reviewer_M21R · 2023-11-23

**Novelty:** 6
**Technical Quality:** 6

**Review:**

The authors have employed a 12-month dataset from X, utilizing toxicity score-based persistence analysis, n-gram-based Multiple Correspondence Analysis (MCA), and topic modeling-based Spearman’s rank correlation. This approach explores the complexities and nuances in anti-Asian hatred narratives across different ethnic communities,  underscores the importance of a disaggregated approach, and offers suggestions on model and technique choices. However,  it could be strengthened by providing more examples and explanations, especially for some "unexpected" results. For example, toxicity peaked during key political events in 2019 rather than the COVID-19 pandemic needs more explanation, including examples to distinguish between racist/hateful content and political criticism. Further, the topics related to K-pop and toxicity need more elaboration.

**Questions:**

1. The manuscript notes that average toxicity peaked not during the COVID-19 pandemic but in August 2019, coinciding with the protests in Hong Kong and the status of Jammu and Kashmir in India. More detailed explanations for this finding are necessary. Presenting examples of tweets classified as toxic during this period would aid in understanding whether these messages are driven by racism or hatred, or if they are primarily anti-Chinese or anti-Indian government sentiments. Clarification is also needed on why topics related to K-pop and its fans are labeled as racist speech, and whether these toxic tweets are influenced by fan culture or actual racist rhetoric.

2. Figure 1c requires further clarification regarding the proportions represented. It is unclear what the baseline group is for the proportion of tweets containing references to each ethnicity.

**Reviewer Confidence:**

4: The reviewer is certain that the evaluation is correct and very familiar with the relevant literature

**Scope:**

4: The work is relevant to the Web and to the track, and is of broad interest to the community

---

### Official Review · Reviewer_QAWA · 2023-11-24

**Novelty:** 3
**Technical Quality:** 5

**Review:**

This paper studies the nuances of anti-Asian racism social media posts between sub-ethnic groups. The authors used several methods to show that languages targeting different sub-ethnic Asian groups are different. The motivation and research gap are valid---most existing work considers a generic anti-Asian theme, but there are lots of nuances within the sub-ethnic Asian groups because of drastically different languages and cultures. ML models trained on generic anti-Asian posts may perform well for the prevailing sub-ethnic groups such as Chinese and Indian but perform poorly for low resource languages such as Vietnamese and Indonesian.

I do like their thorough descriptions about the experiments, how they collected data, how they annotated data, what the inter-rate agreement is, how to build the anti-Asian detection classifier and apply the classifier on the unlabeled data. The outcome of this pipeline is a tweet dataset, each tweet has two labels: targeted sub-ethnic Asian group and anti-Asian or not.

**Pros:**
- Experimental descriptions are clear and results are easy to consume

**Cons:**
- There is not much novelty in this work. All methods are off-the-shelf.
- While the results are clear, I do find it a bit shallow. There is actually just one core message: there are differences within the sub-ethnic Asian groups, supported by evidences from several methods, and future research on detecting anti-Asian racism should consider these nuances and use disaggregated approach.

**Questions:**

See **Cons** above

**Reviewer Confidence:**

4: The reviewer is certain that the evaluation is correct and very familiar with the relevant literature

**Scope:**

4: The work is relevant to the Web and to the track, and is of broad interest to the community

---

### Official Review · Reviewer_f7wm · 2023-11-24

**Novelty:** 2
**Technical Quality:** 6

**Review:**

The authors conducts sub-group analysis to shed light on anti-Asian sentiments on X/Twitter

Pros
* This is an important, but under-studied topic
* While the findings are largely confirming to common expectations, it provides a novel perspective in the context of social media
* The key measurement/detection task is well conducted (classification)
* The sampling strategy (keyword-based) sounds reasonable
* The MCA and BERTopic modeling are concise and revealing

Cons
* Although this is somewhat inevitable, the authors should discuss the potential bias due to their focus on Twitter, particularly in terms of the social/partisan/demographic makeup, and limitations to generalizability

**Questions:**

* More information about the performance of the classifier is necessary such as precision, recall, and F-1 (by class)

**Ethics Review Description:**

-

**Reviewer Confidence:**

4: The reviewer is certain that the evaluation is correct and very familiar with the relevant literature

**Scope:**

4: The work is relevant to the Web and to the track, and is of broad interest to the community

---

### Official Review · Reviewer_z4SY · 2023-12-01

**Novelty:** 6
**Technical Quality:** 5

**Review:**

This paper is well-written and easy to understand. Topic-wise it is very timely, and if the analysis is strong it could be applied to similar contexts for other demographic groups. I believe that focusing on a period surrounding the initial COVID 2019 outbreak is a very good idea as my intuition is that anti-Asian hate was magnified during that period. I would also expect that the differences in hate directed towards different sub-demographics would be easier to see in the data.

Pros:
1. Well written and very easily interpreted.
2. Great topic.
3. Interesting analysis methodology.
4. Well-presented results.

Cons:
1. No ethics or limitation section (for a topic like this I would expect to see both)
2. Data collection may have some biasing issues which I expand upon in the questions
3. Not an egregious issue, but Fig 2 is very difficult to interpret for someone with color-deficient vision.

**Questions:**

I believe the authors are aware of the primary question that a reader would have: "How are you sure that emotional attributes relate to the existence of anti-Asian messaging?" The authors mention in Section 3.2.2 that Perspective API isn't enough and that manual coding was necessary to bolster the training data and improve the strength of the findings but I'm not certain that emotionality should have been a driving measure of this analysis at all. It seems that the samples that were manually coded were driven by stratifications made based on the emotionality scores and I'm not certain that this should have been part of the preprocess.

What was the rationale for using generic keywords to drive your data collection? It seems to me that there was likely a lot of messaging missed by not adding in nonstandard keywords especially given how frequently they are likely used in hateful messaging.

**Ethics Review Description:**

I didn't select yes. I think the paper should have an ethics section to discuss the IRB and data safety issue that are likely present in the work though.

**Reviewer Confidence:**

3: The reviewer is confident but not certain that the evaluation is correct

**Scope:**

4: The work is relevant to the Web and to the track, and is of broad interest to the community

---

### Decision · Program_Chairs · 2024-01-22

**Decision:**

Accept

**Comment:**

Quoting the summary by Reviewer M21R: "The authors have employed a 12-month dataset from X, utilizing toxicity score-based persistence analysis, n-gram-based Multiple Correspondence Analysis (MCA), and topic modeling-based Spearman's rank correlation. This approach explores the complexities and nuances in anti-Asian hatred narratives across different ethnic communities, underscores the importance of a disaggregated approach, and offers suggestions on model and technique choices."

 Overall, there was very strong interest in the topic, as well as agreement that the results are clearly presented, and a great fit for the track/conference. There was also universal agreement that the analysis was well performed. The main contention concerns the novelty, which two reviewers view as sub-average, mostly due to the fact that the methods used are off-the-shelf. However, in my own reading of the paper and the interpretation of the "Social Networks, Social Media, and Society" track, a methodological novelty is not a requirement (though often desirable). There were also minor concerns that the analysis and the results might be too shallow or simple. I agree that the core take-away message of the paper is simple, but I personally still view it as important and, in its breadth, as novel.